# Latent Structured Active Learning

**Wenjie Luo**
TTI Chicago
wenjie.luo@ttic.edu

**Alexander G. Schwing**
ETH Zurich
aschwing@inf.ethz.ch

**Raquel Urtasun**
TTI Chicago
rurtasun@ttic.edu

## Abstract

In this paper we present active learning algorithms in the context of structured prediction problems. To reduce the amount of labeling necessary to learn good models, our algorithms operate with weakly labeled data and we query additional examples based on entropies of local marginals, which are a good surrogate for uncertainty. We demonstrate the effectiveness of our approach in the task of 3D layout prediction from single images, and show that good models are learned when labeling only a handful of random variables. In particular, the same performance as using the full training set can be obtained while only labeling ~10% of the random variables.

## 1 Introduction

Most real-world applications are structured, *i.e.*, they are composed of multiple random variables which are related. For example, in natural language processing, we might be interested in parsing sentences syntactically. In computer vision, we might want to predict the depth of each pixel, or its semantic category. In computational biology, given a sequence of proteins (*e.g.*, lethal and edema factors, protective antigen) we might want to predict the 3D docking of the anthrax toxin. While individual variables could be considered independently, it has been demonstrated that taking relations into account improves prediction performance significantly.

Prediction in structured models is typically performed by maximizing a scoring function over the space of all possible outcomes, an NP-hard task for most graphical models. Traditional learning algorithms for structured problems tackle the supervised setting [16, 33, 11], where input-output pairs are given and each structured output is fully labeled. Obtaining fully labeled examples might, however, be very cumbersome as structured models often involve a large number of random variables, *e.g.*, in semantic segmentation, we have to label several million random variables, one for each pixel. Furthermore, obtaining ground truth is sometimes difficult as it potentially requires accessing extra sensors, *e.g.*, laser scanners in the case of stereo. This is even more extreme in the medical domain, where obtaining extra labels is sometimes not even possible, *e.g.*, when tests are not available. Thus, reducing the amount of labeled examples required for learning the scoring function is key for the success of structured prediction in real-world applications.

The active learning setting is particularly beneficial as it has the potential to considerably reduce the amount of supervision required to learn a good model, by querying only the most informative examples. In the structured case, active learning can be generalized to query only subparts of the graph for each example, reducing the amount of necessary labeling even further.

While a variety of active learning approaches exists for the case of classification and regression, the structured case has been less popular, perhaps because of its intrinsic computational difficulties as we have to deal with exponentially sized output spaces. Existing approaches typically consider the case where exact inference is possible [7], label the full output space [7, 22], or rely on computationally expensive processes that require inference for each possible outcome of each random variable [34]. The latter is computationally infeasible for most graphical models.

In contrast, in this paper we present efficient approximate approaches for general graphical models where exact inference is intractable. In particular, we propose to select which parts to label based on the entropy of the local marginal distributions. Our active learning algorithms exploit recently developed weakly supervised methods for structured prediction [28], showing that we can benefit from unlabeled examples and exploit the marginal distributions computed during learning. Furthermore, computation is re-used at each active learning iteration, improving efficiency significantly. We demonstrate the effectiveness of our approach in the context of 3D room layout estimation from single images, and show that state-of-the-art results are achieved by employing much fewer manual interactions (*i.e.*, labels). In particular, we match the performance of the state-of-the-art in this task [27] while only labeling ∼10% of the random variables.

In the remainder of the paper we first review learning methods for structured prediction. We then propose our active learning algorithms, and show our experimental evaluation followed by a discussion on related work and conclusions.

## 2   Maximum Likelihood Structure Prediction

We begin by reviewing structured prediction approaches that employ both fully labeled training sets as well as those that handle latent variables. Of particular interest to us are probabilistic formulations since we employ entropies of local probability distributions as our criteria for deciding which parts of the graph to label during each active learning step.

Let $x \in \mathcal{X}$ be the input space (*e.g.*, an image or a sentence), and let $s \in \mathcal{S}$ be the structured labeled space that we are interested in predicting (*e.g.*, an image segmentation or a parse tree). We define $\phi : \mathcal{X} \times \mathcal{S} \to \mathbb{R}^F$ to be a mapping from input and label space to an $F$-dimensional feature space. Here we consider log-linear distributions $p_w(s|x)$ describing the probability over a structured label space $\mathcal{S}$ given an object $x \in \mathcal{X}$ as

$$p_w(s|x) \propto \exp\left(w^\top \phi(x,s)\right). \tag{1}$$

During learning, we are interested in estimating the parameters $w \in \mathbb{R}^F$ of the log-linear distribution such that the score $w^\top \phi(x,s)$ is high if $s \in \mathcal{S}$ is a "good" label for $x \in \mathcal{X}$.

### 2.1   Supervised Setting

To define "good," in the supervised setting we are given a training set $\mathcal{D} = \{(x_i, s_i)_{i=1}^N\}$ containing $N$ pairs, each composed of an input $x \in \mathcal{X}$ and some fully labeled data $s \in \mathcal{S}$. In addition, we are often able to compare the fitness of an estimate $\hat{s} \in \mathcal{S}$ for a training sample $(x, s) \in \mathcal{D}$ via what we refer to as the task-loss function $\ell_{(x,s)}(\hat{s})$. Its purpose is very much like enforcing a distance between the hyperplane defined by the parameters and the respective sample when considering the popular max-margin setting. We incorporate this loss function into the learning process by considering the loss-augmented distribution

$$p_{(x,s)}(s|w) \propto \exp(w^\top \phi(x,s) + \ell_{(x,y)}(s)). \tag{2}$$

Intuitively it places more probability mass on those parts of the output space $\mathcal{S}$ that have a high loss, forcing the model to adapt to a more difficult setting than the one encountered at inference, where the loss is not present.

Maximum likelihood learning aims at finding model parameters $w$ which assign highest probability to the training set $\mathcal{D}$. Assuming the data to be independent and identically distributed (i.i.d.), our goal is to minimize the negative log-posterior $-\ln[p(w)\prod_{(x,s)\in\mathcal{D}} p_{(x,s)}(s|w)]$ with $p(w) \propto e^{-\|w\|_p^p}$ being a prior on the model parameters. The cost function is therefore given by

$$\frac{C}{p}\|w\|_p^p + \sum_{(x,y)\in\mathcal{D}} \left( \epsilon \ln \sum_{\hat{s}\in\mathcal{S}} \exp\left(\frac{w^\top \phi(x,\hat{s}) + \ell_{(x,y)}(\hat{s})}{\epsilon}\right) - w^\top \phi(x,s) \right), \tag{3}$$

where we have included a parameter $\epsilon$ to yield a soft-max function. Although being a convex function, the difficulty arises from the sum over exponentially many label configurations $\hat{s}$.

Different algorithms have been proposed to solve this task. While efficient computation over tree-structured models is required for convergence guarantees [16], approximations were suggested to achieve convergence even when working with loopy models [11].

## 2.2 Dealing with Latent Variables

In the weakly supervised setting, we are given a training set $\mathcal{D} = \{(x_i, y_i)_{i=1}^N\}$ containing $N$ pairs, each composed of an input $x \in \mathcal{X}$ and some partially labeled data $y \in \mathcal{Y} \subseteq \mathcal{S}$. For every training pair, the label space $\mathcal{S} = \mathcal{Y} \times \mathcal{H}$ is divided into two non-intersecting subspaces $\mathcal{Y}$ and $\mathcal{H}$. We refer to the missing information $h \in \mathcal{H}$ as hidden or latent. As before, we incorporate a task-loss function, and define the loss-augmented likelihood of a prediction $\hat{y} \in \mathcal{Y}$ when observing the pair $(x, y)$ as

$$p_{(x,y)}(\hat{y}|w) \propto \sum_{\hat{h} \in \mathcal{H}} p_{(x,y)}(\hat{y}, \hat{h}|w) = \sum_{\hat{h} \in \mathcal{H}} p_{(x,y)}(\hat{s}|w), \tag{4}$$

with $p_{(x,y)}(\hat{s}|w)$ defined as in Eq. 2. The minimization of the negative log-posterior results in the difference of two convex terms as follows

$$\frac{C}{p}\|w\|_p^p + \sum_{(x,y) \in \mathcal{D}} \left( \epsilon \ln \sum_{\hat{s} \in \mathcal{S}} \exp\left( \frac{w^\top \phi(x, \hat{s}) + \ell_{(x,y)}(\hat{s})}{\epsilon} \right) - \epsilon \ln \sum_{\hat{h} \in \mathcal{H}} \exp\left( \frac{w^\top \phi(x, y, \hat{h}) + \ell^c_{(x,y)}(y, \hat{h})}{\epsilon} \right) \right),$$

with the first two terms being the sum of the log-prior and the logarithm of the partition function. For generality we allow different task-loss $\ell, \ell^c$ while noting that $\ell^c \equiv 0$ in our experiments.

Besides the previously outlined difficulty of exponentially sized product spaces, the cost function is no longer convex. Hence we generally employ expectation maximization (EM) or concave-convex procedure (CCCP) [37] type of approaches, *i.e.*, we linearize the non-convex part at the current iterate before taking a step in the direction of the gradient of a convex function. More specifically, we follow Schwing *et al.* [28] and upper-bound the concave part via a minimization over a set of dual variables subsequently referred to as $q_{(x,y)}(h)$:

$$\frac{C}{p}\|w\|_p^p + \sum_{(x,y)} \left( \epsilon \ln \sum_{\hat{s} \in \mathcal{S}} \exp\left( \frac{w^\top \phi(x, \hat{s}) + \ell_{(x,y)}(\hat{s})}{\epsilon} \right) - \epsilon H(q_{(x,y)}) - \mathbb{E}_{q_{(x,y)}}[w^\top \phi(x, y, \hat{h}) + \ell^c(x, y, \hat{h})] \right).$$

To deal with the exponential complexity we notice that frequently the $k$-th element of the feature vector decomposes into local terms, *i.e.*, $\phi_k(x, s) = \sum_{i \in V_{k,x}} \phi_{k,i}(x, s_i) + \sum_{\alpha \in E_{k,x}} \phi_{k,\alpha}(x, s_\alpha)$. $V_{k,x}$ represents the set indexing the unary potentials for the $k$-th feature of example $(x, y)$. Similarly $E_{k,x}$ denotes the set of all high-order variable interaction sets $\alpha$ in the $k$-th feature of example $(x, y)$. All variable indexes which are not observed are subsumed within the set $\mathbb{H}$. Similarly all factors $\alpha$ that contain variable $i$ are summarized within the set $N(i)$.

We leverage the decomposition within the features to also approximate the entropy over the joint distribution $q_{(x,y)}(h)$ by local ones ranging over marginals. Furthermore, we approximate the marginal polytope by the local polytope. We deal with the summation over the output space objects $\hat{s} \in S$ in the convex part in a similar manner. To this end we change to the dual space, employ the entropy approximations and transform the resulting surrogate function back to the primal space where we obtain Lagrange multipliers $\lambda$ which enforce the marginalization constraints. Altogether we obtain an approximate primal program having the following form:

$$\min_{d, \lambda, w} \quad f_1(w, d, \lambda) + f_2(d) + f_3(d) \tag{5}$$

$$\text{s.t.} \quad \sum_{h_\alpha \setminus h_i} d_{(x,y),\alpha}(h_\alpha) = d_{(x,y),i}(h_i) \quad \forall (x,y), i \in \mathbb{H}, \alpha \in N(i), h_i \in \mathcal{S}_i$$

$$d_{(x,y),i}, d_{(x,y),\alpha} \in \underline{\Delta}$$

with $\underline{\Delta}$ denoting probability simplexes. We refer the reader to [28] for the specific forms of these functions.

Following EM or CCCP, this program is optimized by alternatively minimizing w.r.t. the local beliefs $d$ to solve the latent variable prediction problem, and performing a gradient step w.r.t. the weights as well as block-coordinate descent steps to update the Lagrange multipliers $\lambda$. The latter is equivalent to solving a supervised conditional random field problem given the distribution over latent variables inferred in the preceding latent variable prediction step.

We augment [28], and return not only the weights but also the local beliefs $d$ which represent the joint distribution $q_{(x,y)}(h)$, *i.e.*, a distribution over the latent space only. We summarize this process in Alg. 1. Note that only a local minimum is obtained as we are solving a non-convex problem.

**Algorithm 1** latent structured prediction

---

    **Input:** data $\mathcal{D}$, initial weights $w$
    **repeat**
      **repeat**
        //solve latent variable prediction problem
        $\min_d f_2 + f_3$ s.t. $\forall (x,y) \; d_{(x,y)} \in \mathcal{D}_{(x,y)}$
      **until** convergence
      //message passing update
      $\forall (x,y), i \in \mathbb{S} \quad \lambda_{(x,y),i} \leftarrow \nabla_{\lambda_{(x,y),i}}(f_1 + f_2) = 0$
      //gradient step with step size $\eta$
      $w \leftarrow w - \eta \nabla_w (f_1 + f_2)$
    **until** convergence
    **Output:** weights $w$, beliefs $d$

---

## 3 Active Learning

In the previous section, we defined the maximum likelihood estimators for learning in the supervised and weakly supervised setting. We now derive our active learning approaches. In the active learning setting, we assume a given training set $\mathcal{D}_S = \{(x_i, y_i)_{i=1}^{N_L}\}$ containing $N_L$ pairs, each composed by an input $x \in \mathcal{X}$ and some partially labeled data $y \in \mathcal{Y} \subseteq \mathcal{S}$. As before, for every training pair, we divide the label space $\mathcal{S} = \mathcal{Y} \times \mathcal{H}$ into two non-intersecting subspaces $\mathcal{Y}$ and $\mathcal{H}$, and refer to the missing information $h \in \mathcal{H}$ as hidden or latent. Additionally, we are given a set of unlabeled examples $\mathcal{D}_U = \{(x_i)_{i=1}^{N_u}\}$.

We are interested in answering the following question: which part of the graph for which example should we labeled in order to learn the best model with the least amount of supervision? Towards this goal, we derive iterative algorithms which select the random variables to be labeled based on the local entropies. This is intuitive, as entropy is a surrogate for uncertainty and useful for the considered application since the cost of labeling a random variable is independent of the selection.

Here, our algorithms iteratively query the labels of the random variables of highest uncertainty, update the model parameters $w$ and again ask for the next most uncertain set of variables.

Towards this goal, we need to compute the entropies of the marginal distributions over each latent variable, as well as the entropy over each random variable of the unlabeled examples. This is in general NP-hard, as we are interested in dealing with graphical models with general potentials and connectivity. In this paper we derive two active learning algorithms, each with a different trade-off between accuracy and computational complexity.

**Separate active:** Our first algorithm utilizes the labeled and weakly labeled examples to learn at each iteration. Once the parameters are learned it performs inference over the unlabeled and partially labeled examples to query for the next random variable to label. Thus, it requires a separate inference step for each active learning iteration. As shown in our experiments, this can be done efficiently using convex belief propagation [10, 26]. The corresponding algorithm is summarized in Alg. 2.

**Joint active:** Our second active learning algorithm takes advantage of unlabeled examples during learning and no extra effort is required to compute the most informative random variable. Note that this contrasts active learning algorithms which typically do not exploit unlabeled data during learning and require very expensive computations in order to select the next example or random variable to be labeled. Let $\mathcal{D}^1 = \mathcal{D}_S \cup \mathcal{D}_U$ be the set of all training examples containing both fully labeled, partially labeled and unlabeled examples. At each iteration we obtain $\mathcal{D}^t$ by querying the label of a random variable not being labeled in $\mathcal{D}^{t-1}$. Thus, at each iteration, we learn using a weakly supervised structured prediction task that solves

$$\frac{C}{p}\|w_t\|_p^p + \sum_{(x,y)\in\mathcal{D}^t}\left(\epsilon \ln \sum_{\hat{s}\in\mathcal{S}} \exp\left(\frac{w_t^\top \phi(x,\hat{s}) + \ell_{(x,y)}(\hat{s})}{\epsilon}\right) - \epsilon \ln \sum_{\hat{h}\in\mathcal{H}^t} \exp\left(\frac{w_t^\top \phi(x,y,\hat{h}) + \ell_{(x,y)}^c(y,\hat{h})}{\epsilon}\right)\right),$$

| **Algorithm 2** Separate active | **Algorithm 3** Joint active |
|---|---|
| **Input:** data $\mathcal{D}_S, \mathcal{D}_U$, initial weights $w$ | **Input:** data $\mathcal{D}_S, \mathcal{D}_U$, initial weights $w$ |
| **repeat** | **repeat** |
| $\quad (w, d_S) \leftarrow$ Alg. 1$(\mathcal{D}_S, w)$ | $\quad (w, d) \leftarrow$ Alg. 1$(\mathcal{D}_S \cup \mathcal{D}_U, w)$ |
| $\quad d_U \leftarrow$ Inference$(\mathcal{D}_U)$ | $\quad i^* \leftarrow \arg\max_i H(d_i)$ |
| $\quad i^* \leftarrow \arg\max_i H(d_i)$ | $\quad \mathcal{D}_S \leftarrow \mathcal{D}_S \cup \{(x_{i^*}, y_{i^*})\}, \mathcal{D}_U \leftarrow \mathcal{D}_U \setminus x_{i^*}$ |
| $\quad \mathcal{D}_S \leftarrow \mathcal{D}_S \cup \{(x_{i^*}, y_{i^*})\}, \mathcal{D}_U \leftarrow \mathcal{D}_U \setminus x_{i^*}$ | **until** sufficiently certain |
| **until** sufficiently certain | **Output:** weights $w$ |
| **Output:** weights $w$ | |

with $w_t$ the weights for the $t$-th iteration. We resort to the approximated problem given in Eq. 5 to solve this optimization task. The entropies are readily computable in close form, as the local beliefs $d$ are computed during learning. Thus, no extra inference step is necessary. The local entropies are then given by $H(d_i) = -\sum_{h_i=1}^{|\mathcal{H}_i|} d_i(h_i) \log d_i(h_i)$, and we query the variable that has the highest entropy, *i.e.*, the highest uncertainty. Note that this computation is linear in the number of unlabeled random variables and linear in the number of states. We summarize our approach in Alg. 3. Note that this algorithm is more expensive than the previous one as learning employs the fully, weakly and unlabeled examples. This is particularly the case when the pool of unlabeled examples is large. However, as shown in our experimental evaluation, it can dramatically reduce the amount of labeling required to learn a good model.

**Batch mode:** The two previously defined active learning approaches are computationally expensive as for each sequential active learning step, a new model has to be learned and inference has to be performed over all latent variables. We also investigate batch algorithms which label $k$ random variables at each step of the algorithm. Towards this goal, we simply label the top $k$ most uncertain variables. Note that this is an approximation of what the sequential algorithm will do, as the estimates of the parameters and the entropies are not updated when selecting the $i$-th variable.

**Re-using computation:** Warm starting the learning algorithm after each active learning query is important in order to reduce the number of iterations required for convergence. Since (almost) the same samples are involved at each step, we can extract a lot of information from previous iterations. To this end we re-use both the weights $w$ as well as the messages $\lambda$ and beliefs. More specifically, for Alg. 2 we first perform inference on only newly selected examples to update the corresponding messages $\lambda$. Only afterwards and together with Lagrange multipliers from the other training images and the current weights, we perform the next iteration and another active step. On the other hand, since we take advantage of all the unlabeled data during the joint active learning algorithm (Alg. 3), we already know the Lagrange multipliers $\lambda$ for every image. Without any further updates we directly start a new active step. In our experimental evaluation we show that this choice results in dramatic speed ups when compared to randomly initializing the weights and messages during every active learning iteration. Note that the joint approach (Alg. 3) requires a larger number of iterations to converge as it employs large amounts of unlabeled data. After a few iterations, convergence for the following active learning steps improves significantly requiring about as much time as the separate approach (Alg. 2) does.

## 4 Experimental Evaluation

We demonstrate the performance of our algorithms on the task of predicting the 3D layout of rooms from a single image. Existing approaches formulate the task as a structured prediction problem focusing on estimating the 3D box which best describes the layout. Taking advantage of the *Manhattan world assumption* (*i.e.*, the existence of three dominant vanishing points which are orthonormal), and given the vanishing points, the problem can be formulated as inference in a pairwise graphical model composed of four random variables [27]. As shown in Fig. 1, these variables represent the angles encoding the rays that originate from the respective vanishing points. Following existing approaches [12, 17], we employ $F = 55$ features based on geometric context (GC) [13] and orientation maps (OM) [18] as image cues. Our features $\phi$ count for each face in the cuboid (given a particular configuration of the layout) the number of pixels with a certain label for OM and the probability that such label exists for GC and the task-loss $\ell$ denotes the pixel-wise prediction error.

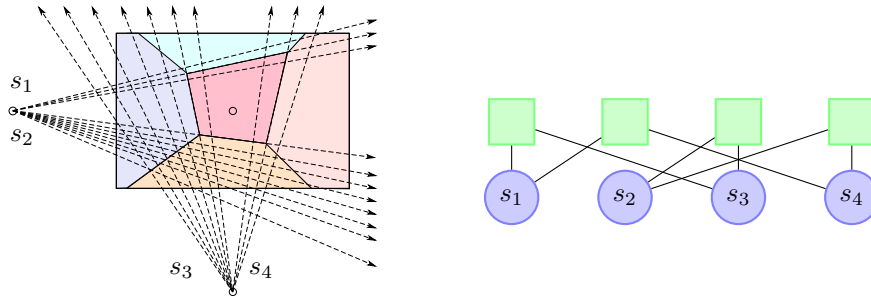

Figure 1: Parameterization and factor graph for the 3D layout prediction task.

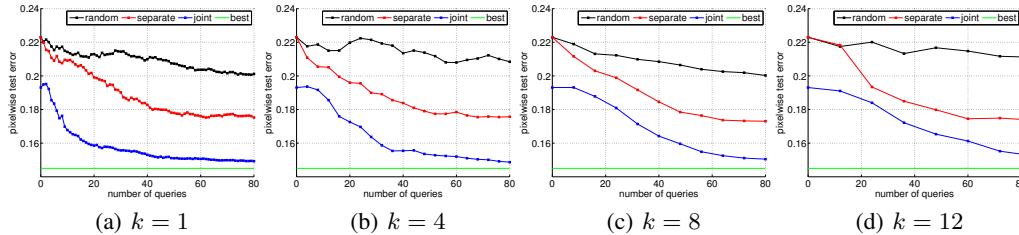

(a) $k = 1$      (b) $k = 4$      (c) $k = 8$      (d) $k = 12$

Figure 2: Test set error as a function of the number of random variables labeled, when using joint vs separate active learning. The different plots reflect scenarios where the top $k$ random variables are labeled at each iteration (*i.e.*, batch setting). From left to right $k = 1, 4, 8$ and 12.

Performance is measured as the percentage of pixels that have been correctly labeled as, left-wall, right-wall, front-wall, ceiling or floor. Unless otherwise stated all experiments are performed by averaging over 20 runs of the algorithm, where the initial seed of 10 fully labeled images is selected at random.

**Active learning:** We begin our experimentation by comparing the two proposed active learning algorithms, *i.e.*, *separate* (Alg. 2) and *joint* (Alg. 3). As shown in Fig. 2(a), both active learning algorithms achieve much lower test error than an algorithm that selects which variables to label at random. Also, note that the *joint* algorithm takes advantage of unlabeled data and achieves good performance after labeling only a few variables, improving significantly over the *separate* algorithm.

**Batch active learning:** Fig. 2 shows the performances of both active learning algorithms when labeling a batch of $k$ random variables before re-learning. Note that even with a batch of $k = 12$ random variables, our algorithms quickly outperform random selection, as illustrated in Fig. 2(d).

**Image vs random variable:** Instead of labeling a random variable at a time, we also experiment with an algorithm that labels the four variables of an image at once. Note that this setting is equivalent to labeling four random variables per image. As shown in Fig. 3(a), labeling the full image requires more labeling to achieve the same test error performance when compared to labeling random variables from possibly different examples.

**Importance of $\epsilon$:** Fig. 3(b) and (c) show the performance of our active learning algorithms as a function of $\epsilon$. Note that this parameter is fairly important. In particular, when $\epsilon = 1$, the entropy of most random variables is too large to be discriminative. This is illustrated in Fig. 3(d) where we observe a fairly uniform distribution over the states of a randomly chosen variable for $\epsilon = 1$. Our active learning algorithm thus prefers smaller values of $\epsilon$. We hypothesize that this is due to the fact that we have a small number of random variables each having a large number of states. Our initial tests show that in other applications where the number of states is smaller (*e.g.*, segmentation) larger values of $\epsilon$ perform better. An automatic selection of $\epsilon$ is subject of our future research.

**Complexity Separate vs. Joint:** In Fig. 4(a) we illustrate the number of CCCP iterations as a function of the number of queried examples for both active learning algorithms. We observe that the *joint* algorithm requires more computation initially. But after the first few active steps, *i.e.*, after having converged to a good solution, its computation requirements reduce drastically. Here we use $\epsilon = 0.01$ for both algorithms.

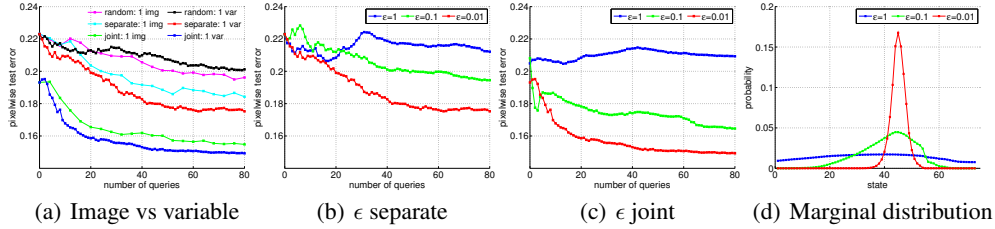

| (a) Image vs variable | (b) $\epsilon$ separate | (c) $\epsilon$ joint | (d) Marginal distribution |

Figure 3: Test set error as a function of the number of random variables labeled ((a)-(c)). Marginal distribution is illustrated in (d) for different $\epsilon$.

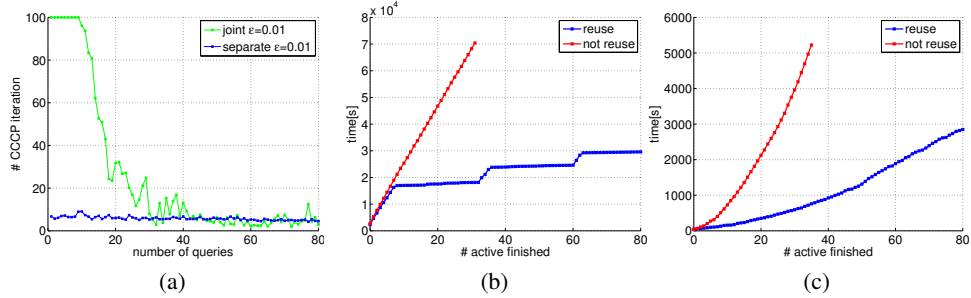

| (a) | (b) | (c) |

Figure 4: Number of CCCP iterations as a function of the amount of queried variables in (a) and time after specified number of active iterations in (b) (joint) and (c) (separate).

**Reusing computation:** Fig. 4(b) and (c) show the number of finished active learning iterations as a function of time for the joint and separate algorithm respectively. Note that by reusing computation, a much larger number of active learning iterations finishes when given a specific time budget.

## 5 Related Work

Active learning approaches consider two different scenarios. In stream-based methods [5], samples are considered successively and a decision is made to discard or eventually pick the currently investigated sample. In contrast, pool-based methods [20] have access to a large set of unlabeled data. Clearly our proposed approach has a pool-based flavor. Over the years many different strategies have been proposed in the context of active learning algorithms to decide which example to label next. While we follow the uncertainty sampling scheme [20, 19] using an entropy measure, sampling schemes based on expected model change [29] have also been proposed. Other alternatives are expected error reduction [24], variance reduction [4, 6], least-confident measure [7] or margin-based measures [25].

An alternative way to classify active learning algorithms is related to the information revealed after querying for a label. In the multi-armed bandit model [1, 2] the algorithm chooses an action/sample and observes the utility of only that action. Alternatively when learning with expert advice, utilities for all possible actions are revealed [3]. Between both of the aforementioned extremes sits the co-active learning setting [30] where a subset of rewards for all possible actions is revealed by the user. Our approach resembles the multi-armed bandit setting since we only get to know the result of the newly queried sample.

Active learning approaches have been proposed in the context of Neural Networks [6], Support Vector Machines [32], Gaussian processes [14], CRFs [7] and structured max-margin formulations [22]. Contrasting many of the previously proposed approaches we consider active learning as an extension of a latent structured prediction setting, *i.e.*, we extend the double-loop algorithm by yet another layer. Importantly, our active learning algorithm follows the recent ideas to unify CRFs and structured SVMs. It employs convex approximations and is amenable to general graphical models with arbitrary topology and energy functions.

The first application of active learning in computer vision was developed by Kapoor *et al*. [14] to perform object recognition with minimal supervision. In the context of structured models [8] proposed to use conditional entropies to decide which image to label next in a segmentation task. In [36] the set of frames to label in a video sequence is selected based on the cost of labeling each frame and the cost of correcting errors. Unlike our approach, [8, 36] labeled full images (not sets of random variables). As shown in our experiments this requires more manual interactions than our approach. GrabCut [23] popularized the use of "active learning" for figure ground segmentation, where the question of what to labeled next is answered by a human via an interactive segmentation system. Siddiquie *et al*. [31] propose to label that variable which most reduces the entropy of the entire "system," *i.e*., all the data, by taking into account correlations between variables using a Bethe entropy approximation. In [15], the next region to be labeled is selected based on a surrogate of uncertainty (*i.e*., min marginals) which is computed efficiently via dynamic graph cuts. This, however, is only suitable for problems that can be solved via graph cuts (*e.g*., binary labeling problems with sub modular energies). In contrast, in this paper we are interested in the general setting of arbitrary energies and connectivities. Entropy was used as an active learning criteria for tree-structured models [21], where marginal probabilities can be computed exactly.

In the context of video segmentation, Fathi *et al*. [9] frame active learning as a semi-supervised learning problem over a graph. They utilized the entropy as a metric for selecting which super-pixel to label within their graph regularization approach. In the context of holistic approaches, Vijayanarasimhan *et al*. [35] investigated the problem of which task to label. Towards this goal they derived a multi-label multiple-instance approach, which takes into account the task effort (*i.e*., the expected time to perform each labeling). Vezhnevets *et al*. [34] resort to the expected change as the criteria to select which parts to label in the graphical model. Unfortunately, computing this measure is computationally expensive, and their approach is only feasible for graphical models where inference can be solved via graph cuts.

# 6 Conclusions

We have proposed active learning algorithms in the context of structure models which utilized local entropies in order to decide which subset of the output space for which example to label. We have demonstrated the effectiveness of our approach in the problem of 3D room layout prediction given a single image, and we showed that state-of-the-art performance can be obtained while only employing $\sim$10% of the labelings. We will release the source code upon acceptance as well as scripts to reproduce all experiments in the paper. In the future, we plan to apply our algorithms in the context of holistic models in order to investigate which tasks are more informative for visual parsing.

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
