[Reviews · NeurIPS 2013]

Submitted by Assigned_Reviewer_3

The authors extend the work of [Schwing et al., Efficient Structured Prediction with Latent Variables for General Graphical Models; ICML 2012] to include active learning protocols. In essence, they use the entropy of the local variable marginals to estimate a value of uncertainty -- so this can be viewed as local variable uncertainty sampling for structured predictions with partially observed variables. To this end, they propose two active learning variants, namely (1) separate active: each active learning round requires a "separate" inference step over the unlabeled and partially labeled variables after learning on the observed variables and (2) joint active: each active learning round follows a joint learning procedure over the labeled, partially labeled, and unlabeled instances. They also explore batch sizes and warm-starting of the learning procedure between rounds in this setting. The empirical evaluation is 3D layout of rooms (a computer vision problem explored in [Scwing et al., ICML 2012], demonstrating that (1) "joint" active learning works the best, achieving an annotation savings of ~90%, (2) batch mode works for reasonable small batches, (3) querying "full" vs. "partial" labels [unsurprisingly, partial labels works better] (4) sensitivity to \epsilon and (5) computational reuse saves time [unsuprisingly].

Overall, it is a nice paper that shows active learning works in a challenging setting with this particular formulation. I will address the {quality, clarity, originality, significance} components separately below.

Quality: The paper is certainly of high quality. The model for structured learning with latent variables is state of the art and the active learning method achieves dramatic savings over just using random sampling (which actually makes quite a bit of sense in the latent variable case when you think about it). My biggest complaint from an active learning perspective is that the saving are somewhat disingenuous (even if not intentionally so) -- while the "$1 for 1 label" simulations were very popular from ~1995-2008ish, I think it has been accepted that these sorts of simulations aren't very realistic since active learning often picks the most difficult examples for labeling (e.g., [Kapoor et al., Selective supervision: Guiding supervised learning with decision-theoretic active learning; IJCAI 2007],[Settles et al., Active learning with real annotation costs; NIPS WS 2008], [Ringger et al., Assessing the costs of machine-assisted corpus annotation through a user study; LREC 2008], [Haertel et al., Return on investment for active learning; NIPS WS 2008], [Arora et al., Estimating annotation cost for active learning in a multi-annotator environment; NAACL WS 2009], [Baldridge and Palmer, How well does active learning actually work? Time-based evaluation of cost-reduction strategies for language documentation; EMNLP 2009], [Wallace et al., Modeling Annotation Time to Reduce Workload in Comparative Effectiveness Reviews; IHI 2010]...amongst others. I think there should at least be some discussion along these lines (although the full vs. partial labels experiments do at least hint at this a bit). My second "active learning" concern is a bit on the semantics side, but both [Culotta and McCallum, AAAI 2005] and [Roth and Small, ECML 2006] consider partial labels during querying time -- which is somewhat in contrast to the last paragraph of page 1. I think what you mean to say is that the output space is fully observable or something along these lines, as they certainly do not require "full" labels from the querying function (and I actually don't think this is what you meant)

Clarity: For the most part, the paper is clear -- although I had to go read [Scwing et al., ICML 2012] to understand some of the details. The biggest thing I noticed in this case is (1) I didn't completely understand the "task-loss" function until I read the previous work; some example/discussion here would be nice (2) I think the notation for this loss function may have suffered from some cutting/pasting from the ICML paper as there appears to be some confusion between s and y here and there (assuming I am understanding everything correctly) and (3) you never say anything about $\ell^c$ -- which may be difficult to digest for those without explicit graphical models expertise.

Originality: While the step is straightforward (and based on uncertainty sampling), it hasn't been done before in this setting. However, I also think anybody with knowledge of these areas could have put these elements together.

Significance: I would guess that this paper will not have the same sort of impact as [Scwing et al., ICML 2012] as it is almost a "companion piece". The ideas are relatively straightforward, but done well. Probably having the code would be more useful than reading the paper as there are no theoretical results or anything along these lines and the setting of $\epsilon$ seems non-trivial in deployed settings. Therefore, I would say it is interesting, but not groundbreaking in this regard.

A few small comments:
abstract: -> $\sim 10\%$ (get rid of the weird spacing)
pg 1: "have to deal with exponentially sized output variables" -> "have to deal with output spaces which are exponentially sized in the number of local variables" (or something along these lines)
pg 2: "that employ both fully labeled training sets" As I previously stated, I think there is a subtle distinction between "labeled" and "observed" -- but this is obviously your call if you agree
pg 5: "as well as the messages $\lambda$ and believes" -> "... beliefs"
Summary: The authors extend the work of [Schwing et al., Efficient Structured Prediction with Latent Variables for General Graphical Models; ICML 2012] to include active learning protocols via entropy of the local variable marginals. It works well empirically on a reasonably difficult task and seems likely to generalize to other similar problems. While there are some small issues that could be cleaned up to make a stronger paper, I think it is a useful result (even if a straightforward extension of [Scwing et al., ICML 2012])

Submitted by Assigned_Reviewer_4

This paper presents an approach to active learning for structured output spaces. The idea is to use local marginals for selection rather than performing inference for all random variables and then selecting the query for active learning. For computing the local marginals, the approach builds upon [28].

I like the direction and the task in the paper. Active learning for structured spaces is an important problem in vision which is hardly being looked at. Therefore, the research in the paper is quite relevant. I think the paper makes some important contribution (although building upon the past work).

However, there are a few issues with the paper:

1. The experiments are shown on indoor geometry task. I believe for broader impact the authors should show experiments on other task such as scene labeling or human pose ...Both of these are structured spaces and will increase the impact hugely.
2. The baselines in the experiment are just ablative ones. I wonder if the paper can be compared to [7],[33]. Also, I am not sure but isn't [7] similar to Algo 1 and [33] similar to Algo2. I would like a more explicit comparison to how the paper is different from [7,33].
3. Sec 2.2 is quite difficult to understand. I would suggest reducing the related work and focusing more on sec 2.2
4. There are absolutely no qualitative results in the paper! I believe showing qualitative results and why one image is preferred over another will help explain the whole active learning process better. I think the paper misses those intuitive discussion about why this approach works ?
5. Finally I think the paper should cite
Siddiquie et al, Beyond Active Noun Tagging: Modeling Contextual Interactions for Multi-Class Active Learning, In CVPR 2010.
This paper performs active learning on structured output space and use the structure/contextual interactions for selecting the queries.
Summary: I like the direction of the paper. While the approach is not completely novel, it takes the right steps and I believe it should be accepted.

Submitted by Assigned_Reviewer_5

This paper presents an active learning approach to structured prediction problems, utilizing an entropy measure to select which nodes in a partially-labeled example to query. The paper builds on recent work by Schwing et al on efficient inference methods for structured learning models with missing labels.

Two variants are proposed and explored. The first finds the output node in unlabelled or partially labeled examples with highest entropy after learning on labeled and partially labelled examples. The second includes the unlabelled examples in learning, and here no extra inference computation is necessary since the entropies for all nodes are computed during learning.

The experiments are clear and results are pretty good. One issue is that the only comparison is to a baseline which selects nodes to label randomly. This is a fairly weak straw-man. It would have been good to explore some other strategy, such as expected error reduction. It also would have been good to examine how the active learning performs as the proportion of weakly labeled to unlabelled to full-labeled data varies. Nonetheless, the consideration of amount of computation required, and effect of the epsilon factor, are well done.

One issue that the authors should address is how approximation errors in entropy (whose estimation relies on convex belief propagation) may affect the results.

Overall the paper is well executed but not especially novel. Uncertainty-based query selection is the standard active learning approach, and the inference and learning methods explored here stay very close to that of Schwing et al. The fact that the results are good are very encouraging, and suggest that this will be a very fruitful direction for structured prediction research.
Summary: This paper presents an active learning approach to structured prediction problems, utilizing an entropy measure to select which nodes in a partially or unlabelled example to query. The method is a fairly straightforward extension of a recent learning/inference method, but the results are pretty strong.
Author Feedback

Author rebuttal: We thank the reviewers for their feedback. We will incorporate all their comments in a revised version of the manuscript.

R1:
- We agree that the $1 for 1 label$ paradigm is not realistic in some applications. However, in the application consider here it is, as it takes an annotator the same cost (i.e., 1 click) to label any of the random variables. We will incorporate a discussion on this in a revised version of the paper.

- We will improve the final version to take the reviewers minor comments into account, as well as to include more details regarding the loss function. In general we followed the notation of [Schwing et al.,ICML 2012], where the joint label space s is decomposed into two disjunct spaces y,h, i.e., s = y \times h.

R2:
- As stated in the submission (l.49ff), [7] operates in a setting where exact inference is tractable. This is not the case in our algorithms, which can work with general graphical models. On the other hand, [33] requires to perform many inference steps, and is thus much more computationally demanding. Our Alg. 2 requires a single inference step for each graphical model, while [33] requires to do inference N*|Y| times for each graphical model in order to compute the expected gain, with N the number of random variables, and |Y| the number of states (assuming for simplicity that all variables have the same number of states). Furthermore, our approach (Alg.3) obtains approximate uncertainty estimates "on the fly" and is therefore inherently suitable for the uncertainty sampling based active learning setting.

- We'll reduce the complexity of the introduced notation in the final version and incorporate more qualitative results as well as the citation mentioned by the reviewer.

R3:
- We will include a comparison to weakly labeled data [Schwing et al.,ICML 2012] and full annotation [Schwing et al.,CVPR 2012] as well as expected error reduction.

- We agree with R3 that approximation errors in the entropy due to the use of convex belief propagation can affect the results. [Meshi et al.,UAI 2009] has some interesting work in this direction. We will add a discussion in the paper.